# Trust as a mediator in the relationship between childhood sexual abuse and IL-6 level in adulthood

Siu-Man Ng[1]*, Ling-Li Leng[1], Qian Wen Xie[1], Jessie S. M. Chan[2,3], Celia H. Y. Chan[1], Kwok Fai So[4,5,6], Ang Li[4], Kevin K. T. Po[5], L. P. Yuen[7], Kam-Shing Ku[8], Anna W. M. Choi[9], Zoë Chouliara[10], Amos C. Y. Cheung[1], Cecilia L. W. Chan[1], Clifton Emery[1]*

1 The Department of Social Work and Social Administration, The University of Hong Kong, sai wan, Hong Kong, 2 Laboratory of Neuropsychology, The University of Hong Kong, sai wan, Hong Kong, 3 Laboratory of Cognitive Affective Neuroscience, The University of Hong Kong, sai wan, Hong Kong, 4 Guangdong-Hong Kong-Macau Institute of CNS Regeneration, Joint International Research Laboratory of CNS Regeneration Ministry of Education, Guangdong Medical Key Laboratory of Brain Function and Diseases, Jinan University, Guangzhou, Guangdong, China, 5 Department of Ophthalmology, LKS Faculty of Medicine, The University of Hong Kong, sai wan, Hong Kong, 6 State Key Laboratory of Brain and Cognitive Science, The University of Hong Kong, sai wan, Hong Kong, 7 International Association for Health and Yangsheng, sai wan, Hong Kong, 8 District Elderly Community Service, Haven of Hope Haven of Hope Christian Service, sai wan, Hong Kong, 9 Department of Social and Behavioral Sciences, City University of Hong Kong, sai wan, Hong Kong, 10 Division of Mental Health and Counselling, Abertay University, Dundee, United kingdom

* ngsiuman@hku.hk (SMN); cemery@hku.hk (CE)

**Data Availability Statement:** We can make the data available in a public repository upon acceptance of the paper.

## Abstract

Childhood sexual abuse (CSA) has been shown to predict the coupling of depression and inflammation in adulthood. Trust within intimate relationships, a core element in marital relations, has been shown to predict positive physical and mental health outcomes, but the mediating role of trust in partners in the association between CSA and inflammation in adulthood requires further study. The present study aimed to examine the impact of CSA on inflammatory biomarkers (IL-6 and IL-1β) in adults with depression and the mediating role of trust. A cross-sectional survey data set of adults presenting with mood and sleep disturbance was used in the analysis. CSA demonstrated a significant negative correlation with IL-6 level ($r = -0.28$, $p < 0.01$) in adults with clinically significant depression, while trust showed a significant positive correlation with IL-6 level ($r = 0.36$, $p < .01$). Sobel test and bootstrapping revealed a significant mediating role for trust between CSA and IL-6 level. CSA and trust in partners were revealed to have significant associations with IL-6 level in adulthood. Counterintuitively, the directions of association were not those expected. Trust played a mediating role between CSA and adulthood levels of IL-6. Plausible explanations for these counterintuitive findings are discussed.

## Introduction

Childhood sexual abuse (CSA) is a global public health problem whose survivors are significantly more vulnerable to both severe mental disorders (e.g., depression, anxiety, and post-

**Funding:** This research was funded by the Innovative Research Fund from the Department of Social Work and Social Administration of The University of Hong Kong.

**Competing interests:** The authors have declared that no competing interests exist.

traumatic stress disorder) and chronic physical diseases (e.g., cancer, diabetes, asthma, and heart disease) in adulthood.[1–5] Prior research indicates that inflammation may be a plausible biological mechanism linking CSA to both physical and mental health problems.[6–10] Inflammation is a protective biological response of immune cells, blood vessels, and molecular mediators to infections or injury. Chronic inflammation develops when this response continues, which may result in disease or even mortality.[11]

There is compelling evidence that childhood trauma is associated with elevated levels of circulating Interleukin inflammatory biomarkers several decades later, such as -6 (IL-6) and Interleukin-1β (IL-1β).[12–18] The association between CSA and inflammatory biomarkers remains unclear due to a significant amount of heterogeneity in the measures and methods employed in the literature.[19–23] Subgroup analyses undertaken in a recent meta-analysis study demonstrate only a weak association between CSA and adulthood IL-6 level.[24]

The relationship between CSA and inflammation may be more complex in individuals with depressive symptoms due to the interplay of three factors: CSA, depression, and inflammation. [25–27] Previous research has found that only severe childhood abuse predicts the coupling of depression and inflammation.[22,28] One recent study suggests that childhood abuse and depression interact to predict IL-6 in pregnant adolescents: more severe childhood abuse and higher levels of depression predicted higher levels of IL-6 than did high abuse and low depression, while less severe childhood abuse and higher levels of depression predicted similar levels of IL-6 to high levels of both abuse and depression.[27]

A close, supportive, happy marital relationship is an important interpersonal resource throughout adulthood, especially when facing difficulties.[29] In the past decade, mounting evidence has shown a positive association between the quality of the marital relationship and physical and mental health outcomes across the adult lifespan.[30–37] High-quality marital relationships significantly predict lower levels of inflammation (including lower IL-6 and IL-1β levels), especially among women.[38–41] Trust, a fundamental factor in marital relationships, is critical to improving intimacy and marital quality, and even marital longevity.[42–43] Recent research has suggested that a high level of interpersonal trust predicts positive physical and mental health outcomes.[38] Trust has also been shown to be central to recovery from CSA in adulthood both within and without mental health services, as well as a sign of recovery. [44–46] Trust has also been shown to be an important factor in survivors' satisfaction with mental health services.[47]

CSA is associated with negative impacts on the stability and quality of intimate relationships in adulthood.[48–51] Adults who experienced CSA may struggle to trust intimate partners. [52–55] A low level of interpersonal trust is associated with negative physical health outcomes through elevated levels of depression.[48] Individuals who feel that it is hard to trust a partner are more likely to experience depression and worse physical health than those who have trusting intimate relationships. However, no research has explored the mediating role of trust in partners in the association between CSA and inflammation in adulthood.

The current study has the following two objectives: (1) to examine the impacts of CSA on inflammatory biomarkers (i.e., IL-6 and IL-1β) in adults with depression; and (2) to explore how marital quality (especially trust in partners) impacts the association between CSA and the pro-inflammatory cytokine network in adulthood.

## Materials and methods

### Participants

The present study utilized data from the baseline assessments of a randomized controlled trial of group therapy for sleep and mood disturbances conducted in 2014. Participants were adults

suffering from poor sleep quality as determined by a score above 5 on the Pittsburgh Sleep Quality Index. The inclusion criteria also included a score between 10 and 34 on the Centre for Epidemiologic Studies Depression Scale, indicating a range from mild depressive symptoms to clinically significant depression. Persons with a history of psychosis or sleep disorders other than insomnia (such as sleep apnea syndrome) were excluded. All participants provided written informed consent. Ethical approval was obtained from the Institutional Review Board of The University of Hong Kong and the West Hong Kong Island Cluster, Hong Kong Hospital Authority (trial registration number: HKCTR-1929 at http://www.hkclinicaltrials.com).

Among the 263 participants of the trial, 194 were living with a spouse or partner at the time of recruitment. These 194 participants comprised the sample for the present study.

## Measures

**Biological markers.** *IL-6 & IL-1β*. Whole blood samples obtained from participants were placed in EDTA tubes (BD Vacutainer PLUS Blood Collection Tubes, BD, Franklin Lakes, NJ) and subjected to centrifugation (1,000 g) at 4˚C for 15 minutes. The supernatants were transferred into new tubes and stored at -80˚C. The plasma levels of IL-6 and IL-1β were measured using the Quantikine HS Human IL-6 and IL-1β/IL-1F2 immunoassay kits (R&D Systems Inc., Minneapolis, MN) following the manufacturer's protocols, as described elsewhere.[56] For IL-1β, the inter-assay coefficient of variation (CV) was 8.1% and the intra-assay CV 3.6%; for IL-6, the inter- assay CV was 7.8% and the intra-assay CV 7.4%.

**Self-report scales.** *Childhood Trauma Questionnaire* (CTQ).[57] The CTQ was utilized to assess exposure to five types of childhood maltreatment: 1) emotional abuse (EA), 2) physical abuse (PA), 3) sexual abuse (SA), 4) emotional neglect (EN), and 5) physical neglect (PN). The scale comprises 25 items, five for each trauma experience. Responses are made on a 5-point Likert scale in terms of frequency of occurrence, ranging from 1 (never) to 5 (very often). The Chinese version of the CTQ has demonstrated good reliability and validity in a Chinese population.[58] Scoring cut-offs were also employed in line with the questionnaire manual to categorize the level of each type of childhood maltreatment. Participants with moderate and high severity levels of childhood maltreatment were identified as having a history of that type of childhood maltreatment experience.

*Trust Scale* (TS).[59] The 17-item TS was used to measure participants' level of trust in their partner. The scale consists of three subscales: 1) predictability, which is perceived stability and consistence of one's partner's specific behaviors according to past experience; 2) dependability, level of confidence in one's partner in terms of their reliability in the face of risk and potential hurt; and 3) faith, level of confidence in one's relationship, as well as one's partner's responsiveness and caring when facing an uncertain future. The respondents' responded to the items of the scale by rating their perception of the trustworthiness of their partner on a 7-point scale ranging from -3 (strongly disagree), through 0 (neutral), to 3 (strongly agree).

*Hospital Anxiety and Depression Scale* (HADS).[60] The HADS was utilized to assess participants' level of anxiety and depression symptoms. It is a 14-item measure consisting of seven items each in depression and anxiety subscales. Items are assessed based on the frequency and intensity of symptom occurrence on a 4-point Likert scale, responses ranging from 0 (not at all) to 3 (nearly all the time/definitely as much). Clinically significant depression is quantified by adopting a cut-off score of 6+. This cut-off was suggested by a previous psychometric study on HADS among primary care patients in Hong Kong, with a sensitivity of 78% and specificity of 91%.[61]

*Perceived Stress Scale, 10-item* (PSS-10).[62] The PSS-10 was utilized to measure psychological stress experienced during the previous month.[62] Items of the scale are assessed for

frequency of occurrence on a 5-point Likert scale, with responses ranging from 0 (never) to 4 (very often). The Chinese version of the PSS-10 has demonstrated good reliability, with Cronbach's alpha values between .70 and .83 in Chinese samples.[63,64]

*Somatic Symptoms Inventory* (SSI).[65] The 28-item SSI was utilized to measure self-reported painful and non-painful somatic symptoms experienced in the previous week. Each item is rated on a 5-point Likert scale based on the extent to which each symptom has bothered the respondent over the past week. Scores range from 1 (not at all) to 5 (a great deal).[66]

**Statistical analyses.** Statistical analyses were conducted using Rstudio (MacOS version 1.1.423). Due to the skewed distribution of the IL-6 and IL-1β data, natural logarithms were used to transform the raw data.

First, a series of bivariate Pearson's correlations was computed to determine the relationships between both IL-6 and IL-1β and childhood trauma experiences, psychological outcomes (trust, perceived stress, depression, and anxiety), and physical outcomes (somatic symptoms) for all samples. In parallel, a similar series of bivariate Pearson's correlations was conducted within subgroups of participants with and without clinically significant depression.

Second, according to the results of the bivariate Pearson's correlations, the variables showed significant correlations when log IL-6 and log IL-1β were entered into hierarchical multiple regression analysis as predictors, controlling for the effect of socio-demographics. We also compared different regression models using analysis of variance (ANOVA) to examine the magnitude of the added value of each predictor.

Third, because the suppression effect of trust on sexual abuse in affecting IL-6 was observed in the regression models among participants with clinically significant depression, we also performed mediation analysis to further assess the relationships, using Sobel and bootstrapping tests. All statistical differences were considered significant at $p<0.05$.

## Results

### Participants characteristics

The demographic data of the 194 adults with insomnia in the present study are summarized in Table 1. The average age of the participants was about 56 years, about three-quarters of the participants were female, one-third were educated to college level or above, and two-fifths were working full- or part-time. Approximately 11% of the participants reported a history of CSA.

### Bivariate correlations between log IL-6 and log IL-1β and psychological and physical outcomes

The full sample showed significantly positive correlations between log IL-6 and TS total score ($r = 0.19$, $N = 86$, $p<0.05$) and its Dependability subscale ($r = 0.20$, $N = 86$, $p<0.05$). No variable showed significant correlations with log IL-1β. Table 2 presents correlations.

When the full sample was broken down into two subgroups according to the HADS depression cut-off score, the bivariate correlation results were different (see Table 3). Among the participants with depression, TS total scores remained significantly positively correlated with log IL-6, although the correlation was stronger ($r = 0.36$, $N = 86$, $p<0.01$). The association between log IL-6 and TS total score became non-significant ($r = 0.01$, $N = 62$, $p>0.05$) in those without depression. Apart from TS total score, CSA was also found to be significantly negatively associated with log IL-6 among participants with depression ($r = -0.28$, $N = 86$, $p<0.01$).

**Table 1. Socio-demographic & clinical characteristics of participants in the study.**

| Variables | N | Mean (SD) | n (%) |
|---|---|---|---|
| Age (Years) | 194 | 56.3(8.1) | |
| Female | 194 | | 139(71.6%) |
| Male | 194 | | 55(28.4%) |
| Employment | | | |
| Full-time | 194 | | 53(27%) |
| Part-time | 194 | | 22(11%) |
| Retired | 194 | | 48(25%) |
| Homemaker | 194 | | 65(34%) |
| Unemployed | 194 | | 6(3%) |
| Education Level | | | |
| Primary school | 194 | | 53(27%) |
| Middle school | 194 | | 22(11%) |
| High school | 194 | | 48(25%) |
| College or above | 194 | | 65(34%) |
| Childhood trauma total | 194 | 45.7(14.6) | |
| Physical neglect | 194 | 9.6(4.0) | 85(43.8%) |
| Emotional neglect | 194 | 12.7(5.3) | 73(37.6%) |
| Sexual abuse | 194 | 6.0(2.4) | 21(10.8%) |
| Physical abuse | 194 | 8.2(3.7) | 85(43.8%) |
| Emotional abuse | 194 | 9.3(4.1) | 73(37.6%) |
| Trust scale total | 194 | 12.6(17.2) | |
| Dependability | 194 | 4.2(6.4) | |
| Faith | 194 | 7.5(9.0) | |
| Predictability | 194 | 1.8(5.0) | |
| Perceived stress | 194 | 20.7(4.0) | |
| Depression | 194 | 9.1(3.3) | |
| Anxiety | 194 | 9.1(3.3) | |
| Somatic symptoms | 194 | 62.7(19.2) | |
| Log IL-6 | 148 | .22(.6) | |
| Log IL-β | 83 | -2.6(1.6) | |

## Predictors of Log IL-6

TS total score was entered into the hierarchical multiple regression analysis as the predictor for log IL-6 in the full sample. When socio-demographics, namely age, gender, occupation, and education, were controlled, TS total score significantly predicted log IL-6 ($\beta = 0.18$, $t(142) = 2.2$, $p < 0.05$), and explained a significant proportion of variance in log IL-6 [$R^2 = 0.03$, F(5,142) = 2.9, $p = 0.02$].

In the subsample of participants with clinically significant depression, socio-demographics, CSA, and TS total were entered into the regression model in sequence, generating three regression models (see Table 4). The model (Model 3) in which CSA and TS total were entered as the predictors explained a significant proportion of variance in log IL-6, controlling for the effect of socio-demographics [$R^2 = 0.18$, $R^2_{adjusted} = 0.12$, F(6,79) = 2.94, $p < 0.05$]. Like the result for the full sample, in this model TS total score also significantly predicted log IL-6 ($\beta = 0.32$, $t(79) = 2.8$, $p < 0.01$).

It is also interesting to note that both CSA and TS total score showed a significant contribution when entered into the regression model [sexual abuse: $R^2$ change = 0.06, F(1,74) = 5.1,

**Table 2. Bivariate Pearson's correlations table between IL-6 and IL-1β with childhood trauma experiences, and psychological outcomes physical outcomes for total sample.**

| Variables | Log IL-6 | Log IL-1β |
|---|---|---|
| | N = 148 | N = 83 |
| 1. Childhood trauma total | 0.03 | -0.07 |
| 1.1 Physical neglect | 0.05 | -0.08 |
| 1.2 Emotion neglect | 0.02 | -0.01 |
| 1.3 Sexual abuse | -0.15 | -0.09 |
| 1.4 Physical abuse | 0.06 | -0.10 |
| 1.5 Emotional abuse | 0.05 | -0.11 |
| 2. Trust scale total | 0.19* | 0.06 |
| 2.1 Dependability | 0.20* | 0.11 |
| 2.2 Faith | 0.15 | 0.01 |
| 2.3 Predictability | 0.16 | 0.04 |
| 3. Perceived stress | -0.07 | 0.11 |
| 4. Depression | -0.11 | 0.05 |
| 5. Anxiety | -0.07 | -0.03 |
| 6. Somatic symptoms | -0.05 | -0.02 |

* Correlation is significant at the 0.05 level (2-tailed)

$p < 0.05$; TS total score: $R^2$ change = 0.08, $F(1,73) = 8.0$, $p < 0.01$]. Although CSA significantly predicted log IL-6 ($\beta = -0.25$, $t(74) = -2.3$, $p < 0.05$), it became non-significant after TS total score was entered into the model ($\beta = -0.13$, $t(73) = -1.1$, $p = 0.27$). This might suggest a poten-

**Table 3. Bivariate Pearson's correlations table between IL-6 and IL-1β with childhood trauma experiences, psychological outcomes, and physical outcomes among participants with and without clinically significant depression.**

| Variables | Log IL-6 | | Log IL-1β | |
|---|---|---|---|---|
| | Depressed | Non-depressed | Depressed | Non-depressed |
| | N = 86 | N = 62 | N = 50 | N = 33 |
| 1. Childhood trauma total | -0.09 | 0.20 | -0.04 | -0.24 |
| 1.1 Physical neglect | -0.01 | 0.14 | -0.00 | -0.18 |
| 1.2 Emotion neglect | -0.08 | 0.14 | 0.02 | -0.25 |
| 1.3 Sexual abuse | -0.28** | 0.10 | 0.02 | -0.05 |
| 1.4 Physical abuse | 0.05 | 0.09 | -0.06 | -0.16 |
| 1.5 Emotional abuse | -0.08 | 0.22 | -0.08 | -0.14 |
| 2. Trust scale total | 0.36** | 0.01 | 0.12 | -0.01 |
| 2.1 Dependability | 0.35** | 0.02 | 0.19 | 0.02 |
| 2.2 Faith | 0.35** | -0.05 | 0.05 | -0.02 |
| 2.3 Predictability | 0.23* | 0.08 | 0.10 | -0.02 |
| 3. Perceived stress | -0.10 | -0.02 | 0.21 | -0.05 |
| 4. Depression | -0.16 | -0.03 | 0.03 | 0.01 |
| 5. Anxiety | -0.05 | -0.05 | 0.03 | -0.14 |
| 6. Somatic symptoms | -0.04 | -0.03 | -0.01 | -0.09 |

* Correlation is significant at the 0.05 level (2-tailed)

** Correlation is significant at the 0.01 level (2-tailed)

**Table 4. Hierarchical multiple regression analysis predicting Log IL-6 among participants with clinically significant depression Log IL-6 (N = 86).**

| Predictors | Model 1 | | | Model 2 | | | Model 3 | | |
|---|---|---|---|---|---|---|---|---|---|
| | B | β | p | B | β | p | B | β | p |
| Block 1: Socio-Demographics | | | | | | | | | |
| Age | -0.00 | -0.01 | 0.92 | -0.00 | -0.02 | 0.90 | 0.00 | 0.01 | 0.94 |
| Gender | -0.17 | -0.15 | 0.22 | -0.11 | -0.10 | 0.44 | -0.10 | -0.03 | 0.45 |
| Occupation | 0.06 | 0.16 | 0.19 | 0.04 | 0.09 | 0.43 | 0.03 | 0.07 | 0.56 |
| Education | -0.05 | -0.12 | 0.35 | -0.05 | -0.12 | 0.33 | -0.07 | -0.17 | 0.16 |
| Block 2: | | | | | | | | | |
| Child sexual abuse | | | | -0.05 | -0.25 | 0.03* | -0.02 | -0.13 | 0.27 |
| Block 3: | | | | | | | | | |
| Trust scale total | | | | | | | 0.01 | 0.32 | 0.01** |
| $R^2$ | 0.04; $p > 0.05$ | | | 0.10; $p = 0.13$ | | | 0.18; $p < 0.05$ | | |
| R2 adjusted | -0.001 | | | 0.04 | | | 0.12 | | |
| $R^2$ change | 0.04; $p > 0.05$ | | | 0.06; $p < 0.05$ | | | 0.08; $p < 0.01$ | | |

* Correlation is significant at the 0.05 level (2-tailed)

** Correlation is significant at the 0.01 level (2-tailed)

tial mediation effect of TS total score on CSA in predicting log IL-6, which we assessed in the next step using the Sobel test.

## Mediation analysis: Trust mediates the relationship between child sexual and Log IL-6

The Sobel test indicated that TS total score was a significant mediator of the influence of CSA on log IL-6 level among participants with depression ($z = -2.2$, $p = 0.02$). As Fig 1 illustrates, the standardized regression coefficient between CSA and TS total score was statistically significant [a path: β = -2.28, $t = -4.0$, $p < 0.001$); $F_{(1,84)} = 0.15$, $R^2 = 0.16$, $R^2$ adjusted = 0.15, $p < 0.001$], as was the standardized regression coefficient between TS total score and log IL-6 [b path: β = 0.29, $t = 2.6$, $p < 0.05$); $F_{(2,83)} = 7.3$, $R^2 = 0.15$, $R^2$ adjusted = 0.13, $p < 0.001$]. We tested the significance of this indirect effect using bootstrapping. Unstandardized indirect effects were computed for 1,000 bootstrapped samples, and the 95% confidence interval was computed by determining the indirect effects at the 2.5th and 97.5th percentiles. The bootstrapped unstandardized indirect effect was -0.02 ($p < 0.05$), and the 95% confidence interval ranged from -0.05 to -0.002. Thus, the indirect effect was statistically significant. This result suggests that CSA has a negative influence on the level of IL-6 through the level of trust in one's partner.

## Discussion

Among the five types of childhood trauma measured in this study, CSA showed the strongest association with IL-6 level in participants with clinically significant depression ($r = -0.28$, $p < 0.01$). The other four types of childhood trauma, emotional abuse, physical abuse, emotional neglect, and physical neglect, show statistically non-significant correlations. On one hand, this is understandable because CSA is considered to be the most traumatic of these childhood experiences, and can have stronger long-term negative impacts.[4,15] CSA also seems to coexist with many other types of abuse (e.g. emotional, physical, domestic, neglect, etc.).[3,13] On the other hand, the correlation between CSA and IL-6 level in adulthood found here takes an unexpected direction. It is generally believed that CSA is associated with elevated levels of circulating inflammatory biomarkers.[12–18] The present study reveals a negative

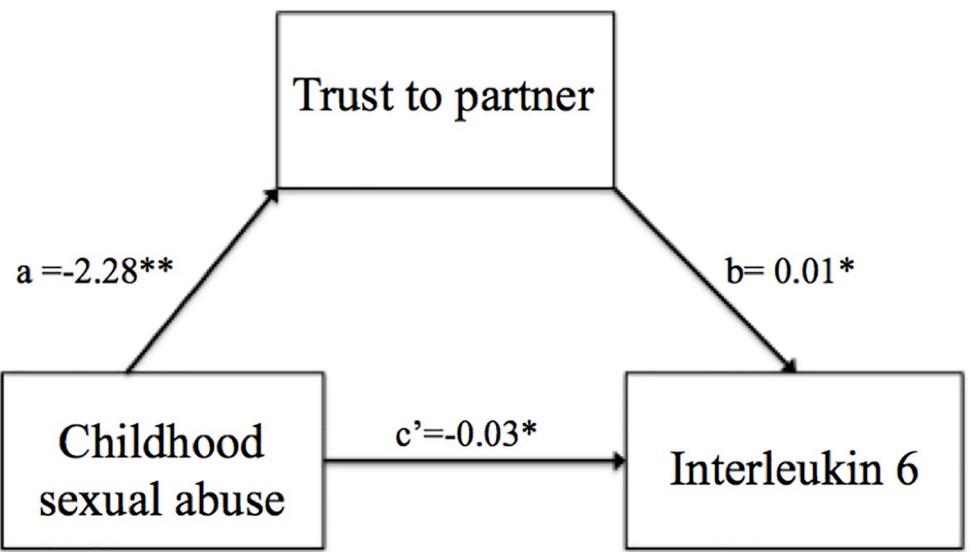

**Fig 1. Mediation model: Trust in partner mediated the relationship of child sexual abuse in affecting IL-6 (N = 86).** * Correlation is significant at the 0.05 level (2-tailed).

correlation between CSA and adulthood IL-6. While there are few studies in this area, a recent meta-analysis reports that the association between CSA and adulthood IL-6 is mixed and complex in depressed patients.[19] The mechanisms of the impacts of CSA on IL-6 or chronic inflammation in adulthood are not entirely clear. Inflammation itself is a very complex, dynamic process in which the level of IL-6 is subject to the influence of many variables, including the levels of other proinflammatory cytokines, such as TNF-α and IL-1β, and the levels of anti-inflammatory cytokines, such as IL-4, IL-10, and IL-11.[67,68] Moreover, there is a negative feedback loop from anti-inflammatory to proinflammatory cytokines.[69] In an inflammatory condition, the level of a single proinflammatory cytokine, such as IL-6, is not necessarily elevated. To estimate the severity of inflammation more accurately, the levels of other proinflammatory and anti-inflammatory cytokines must be evaluated at the same time.

In the present study, trust in partner shows significant association with IL-6 level in participants with clinically significant depression ($r = 0.36$, $p<0.01$). However, the correlation between trust in partner and IL-6 level is also in an unexpected direction. It is generally believed that trust in partner has a negative correlation with IL-6 level.[51–54] The reasons for the observed discrepancy (a positive correlation) are likely similar to those depicted in the above paragraph.

Hierarchical multiple regression analysis showed CSA to be a significant predictor of adulthood IL-6 level. When trust in partner was entered into the model, CSA and trust in partner together explained 12% of variance in IL-6 level in adulthood. It is worth noting that the standardized β of CSA was reduced by a magnitude of 48%, from -0.25 to -0.13. Subsequent mediation analysis suggested that trust in partner mediates the impacts of CSA on adulthood IL-6 level. The findings seem to suggest that trust in partner may mitigate the negative impacts of CSA. Rigorous investigation of this direction is worth pursuing.

The above findings were not observed in participants who were not clinically depressed. CAS did not show significant association with IL-6 level. Previous studies have revealed that CAS had association with depression in adulthood.[3]A plausible explanation for this may be that these participants had largely recovered from CSA. Their physical and mental health conditions were more subject to the influence of other more recent psychosocial factors.

The present study has a number of limitations. First, because a cross-sectional design was adopted, and a non-random sample was utilized in the analysis, the causal relationships among the variables were inferred by statistical analysis only. Second, CSA was measured by recall, which is inherently subject to various forms of bias. For adult participants, CSA can be a rather remote life event. It can be difficult to recall such experiences accurately. Besides, CSA is by nature very traumatic. It is common for survivors to develop defence mechanisms to mitigate their immense suffering, which impedes recall of CSA in adulthood. Third, due to resource constraints, only IL-6 and IL-1β were evaluated in the present study. Other proinflammatory and anti-inflammatory cytokines have not been assessed.

To conclude, the findings of the present study suggest that an individual's trust in their partner may mediate the impacts of CSA on adulthood IL-6 level, which has not been revealed by previous research. This result is conceptually coherent and potentially has important implications for practice. It is worth pursuing further rigorous investigation in this direction.

## Acknowledgments

We would like to thank the participants, the volunteers who helped in the data collection, the volunteers from the International Association for Health and Yangsheng in Hong Kong, and the staff of the Centre on Behavioral Health at The University of Hong Kong, who made this project possible.

## Author Contributions

**Conceptualization:** Siu-Man Ng, Kwok Fai So, L. P. Yuen, Anna W. M. Choi, Clifton Emery.

**Data curation:** Ling-Li Leng, Jessie S. M. Chan.

**Formal analysis:** Ling-Li Leng.

**Investigation:** Jessie S. M. Chan, Celia H. Y. Chan, Ang Li, Amos C. Y. Cheung, Cecilia L. W. Chan.

**Project administration:** Jessie S. M. Chan, Kevin K. T. Po, Kam-Shing Ku.

**Resources:** Ang Li, L. P. Yuen, Cecilia L. W. Chan.

**Supervision:** Siu-Man Ng, Celia H. Y. Chan, Kwok Fai So, Anna W. M. Choi, Zoë Chouliara, Cecilia L. W. Chan, Clifton Emery.

**Writing – original draft:** Siu-Man Ng, Ling-Li Leng, Qian Wen Xie.

**Writing – review & editing:** Siu-Man Ng, Clifton Emery.

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
