## [Decision Letter · Decision Letter 0]

30 Mar 2020

PONE-D-20-00707

Trust as a mediator in the relationship between childhood sexual abuse and IL-6 level in adulthood

PLOS ONE

Dear Ms Leng,

Thank you for submitting your manuscript to PLOS ONE. After careful consideration, we feel that it has merit but does not fully meet PLOS ONE’s publication criteria as it currently stands. Therefore, we invite you to submit a revised version of the manuscript that addresses the points raised during the review process.

We would appreciate receiving your revised manuscript by May 14 2020 11:59PM. To enhance the reproducibility of your results, we recommend that if applicable you deposit your laboratory protocols in protocols.io, where a protocol can be assigned its own identifier (DOI) such that it can be cited independently in the future. For instructions see: http://journals.plos.org/plosone/s/submission-guidelines#loc-laboratory-protocols

We look forward to receiving your revised manuscript.

Kind regards,

Geilson Lima Santana, M.D., Ph.D.

Academic Editor

PLOS ONE

Additional Editor Comments (if provided):

Dear authors,

I believe it is important to adhere to PlosOne's editorial guidelines regarding:

1. Numbering of pages and lines - https://journals.plos.org/plosone/s/submission-guidelines

2. Tables and Tables citations - https://journals.plos.org/plosone/s/file?id=80c1/PLOSOne_formatting_sample_main_body.pdf

The tables must be included in the manuscript and I couldn't find them while reading it. Please, see how to do it in the above url.

3. Data availability: you've said that data was available within the manuscript and in the supplement material. I couldn't find it. Please, read these lines from https://journals.plos.org/plosone/s/submit-now

Data availability statement

Answer the following questions to construct your Data Availability statement. This information will appear in the article, if accepted.

Confirm whether all data reported in the manuscript are publicly available. PLOS requires that authors deposit all reported data and related metadata underlying the study findings in an appropriate public repository, unless already provided in the submission. See the data reporting guidelines.

Describe where the data can be found in full sentences. Use the in-system instructions to draft a suitable statement.

Check the boxes to specify if the data will be available in a repository upon acceptance or if you need journal assistance to make it available. Journal staff will follow up to help later on in the process.

Journal Requirements:

2. Please include a copy of Table 1-4 which you refer to in your text on page 4 and 7.

Reviewers' comments:

Reviewer's Responses to Questions

**Comments to the Author**

1. Is the manuscript technically sound, and do the data support the conclusions?

Reviewer #1: Yes

2. Has the statistical analysis been performed appropriately and rigorously? 

Reviewer #1: Yes

3. Have the authors made all data underlying the findings in their manuscript fully available?

Reviewer #1: Yes

4. Is the manuscript presented in an intelligible fashion and written in standard English?

Reviewer #1: Yes

5. Review Comments to the Author

Reviewer #1: This is an interesting manuscript that further demonstrates the complex relationship of inflammatory markers and childhood abuse in the adult years.

I suggest a few minor edits.

1- Please add all leading zeroes in numbers (0.01 vs .01).

2- Include tables showing the statistical outcomes of other measures (trauma, emotional abuse, etc).

3- Include reference for the following statement "CSA also seems to coexist with many other

types of abuse (e.g. emotional, physical, domestic, neglect, etc.)."

4- Include reference for the following statement "On one hand, this is understandable because CSA is

considered to be the most traumatic of these childhood experiences, and can have

stronger long-term negative impacts."

5- Unless you have data relating to this statement: "At their current stage of personal

development, CSA had only a trivial impact on these participants" do not include.

6- The discussion states that "The above findings were not observed in participants who were not clinically

depressed" Please elaborate more on this, explaining what the results were in those who were not clinically depressed. Include additional information relating to the role of IL6 and depression.

7- Update figure one to read "Trust IN partner" not "Trust TO partner" as per the figure description

6. PLOS authors have the option to publish the peer review history of their article (what does this mean?). If published, this will include your full peer review and any attached files.

Reviewer #1: Yes: Kayla A Chase

---

## [Author Response · Author response to Decision Letter 0]

5 Apr 2020

Dear Editor and Reviewer, 

We thank you for your kind consideration and thoughtful feedbacks to our manuscript! Here are our responses to your comments, which are underlined.

To Editor:

1. Numbering of pages and lines – We added the pages and lines

2. Tables and Tables citations – Tables are included into the manuscript

3. Data availability – Yes, we can make the data available in a repository upon acceptance.

Journal Requirements:

1. Please ensure that your manuscript meets PLOS ONE's style requirements – Yes, we adjusted the format in accordance with PLOS ONE’s style requirements.

2. Please include a copy of Table 1-4 which you refer to in your text on page 4 and 7. – We included the tables.

To reviewer:

`

1- Please add all leading zeroes in numbers (0.01 vs .01). – Yes, we did. 

2- Include tables showing the statistical outcomes of other measures (trauma, emotional abuse, etc). – Yes, we did. 

3- Include reference for the following statement "CSA also seems to coexist with many other types of abuse (e.g. emotional, physical, domestic, neglect, etc.)."- – Yes, we did.

4- Include reference for the following statement "On one hand, this is understandable because CSA is considered to be the most traumatic of these childhood experiences, and can have stronger long-term negative impacts."- – Yes, we did.

5- Unless you have data relating to this statement: "At their current stage of personal development, CSA had only a trivial impact on these participants" do not include. – We deleted this statement. 

6- The discussion states that "The above findings were not observed in participants who were not clinically depressed" Please elaborate more on this, explaining what the results were in those who were not clinically depressed. Include additional information relating to the role of IL6 and depression. – We added the elaboration. 

7- Update figure one to read "Trust IN partner" not "Trust TO partner" as per the figure description – Yes, we revised the figure caption. 

We thank you again for your time.

---

## [Decision Letter · Decision Letter 1]

27 Apr 2020

Trust as a mediator in the relationship between childhood sexual abuse and IL-6 level in adulthood

PONE-D-20-00707R1

Dear Dr. Leng,

We are pleased to inform you that your manuscript has been judged scientifically suitable for publication and will be formally accepted for publication once it complies with all outstanding technical requirements.

With kind regards,

Geilson Lima Santana, M.D., Ph.D.

Academic Editor

PLOS ONE

Additional Editor Comments (optional):

Reviewers' comments:

Reviewer's Responses to Questions

**Comments to the Author**

1. If the authors have adequately addressed your comments raised in a previous round of review and you feel that this manuscript is now acceptable for publication, you may indicate that here to bypass the “Comments to the Author” section, enter your conflict of interest statement in the “Confidential to Editor” section, and submit your "Accept" recommendation.

Reviewer #1: All comments have been addressed

2. Is the manuscript technically sound, and do the data support the conclusions?

Reviewer #1: Yes

3. Has the statistical analysis been performed appropriately and rigorously? 

Reviewer #1: Yes

4. Have the authors made all data underlying the findings in their manuscript fully available?

Reviewer #1: Yes

5. Is the manuscript presented in an intelligible fashion and written in standard English?

Reviewer #1: Yes

6. Review Comments to the Author

Reviewer #1: All comments were addressed, I have no further concerns. I have no concerns about dual publication.

7. PLOS authors have the option to publish the peer review history of their article (what does this mean?). If published, this will include your full peer review and any attached files.

Reviewer #1: Yes: Kayla A. Chae

---

## [Editor Report · Acceptance letter]

4 May 2020

PONE-D-20-00707R1 

Trust as a mediator in the relationship between childhood sexual abuse and IL-6 level in adulthood 

Dear Dr. Leng:

I am pleased to inform you that your manuscript has been deemed suitable for publication in PLOS ONE. Congratulations! Your manuscript is now with our production department. 

With kind regards,

on behalf of

Dr. Geilson Lima Santana 

Academic Editor

PLOS ONE